

# Use of digital strategies in the diagnosis of oral squamous cell carcinoma: a scoping review

Rebeka Thiara Nascimento dos Santos[1], Caroline Augusta Belo Faria[1], Nathalya Fedechen Martins[2], Luiz Gustavo de Sousa Duda Júnior[1], Ana Beatriz Fernandes Azevêdo[1], Weslay Rodrigues da Silva[1] and Ana Paula Veras Sobral[1]

[1] Department of Oral and Maxillofacial Pathology, University of Pernambuco, Recife, Pernambuco, Brazil
[2] Faculty of Pharmacy, Dentistry and Nursing, Federal University of Ceará, Fortaleza, Ceará, Brazil

## ABSTRACT

Telediagnosis uses information and communication technologies to support diagnosis, shortening geographical distances. It helps make decisions about various oral lesions. The objective of this scoping review was to map the existing literature on digital strategies to assist in the diagnosis of oral squamous cell carcinoma. this review was structured based on the 5-stage methodology proposed by Arksey and O'Malley, the Joanna Briggs Institute Manual for Evidence Synthesis and followed the Preferred Reporting Items for Systematic Reviews and Meta-Analyses extension for scoping reviews. The methods were registered on the Open Science Framework. The research question was: What digital strategies have been used to assist in the diagnosis of oral squamous cell carcinoma? The search was conducted on PubMed/MEDLINE, Scopus, Web of Science, Embase, and ScienceDirect. Inclusion criteria comprised studies on telediagnosis, teleconsultation or teleconsultation mediated by a professional and studies in English, without date restrictions. The search conducted in June 2023 yielded 1,798 articles, from which 16 studies were included. Telediagnosis was reported in nine studies, involving data screening through applications, clinical images from digital cameras, mobile phones or artificial intelligence. Histopathological images were reported in four studies. Both, telediagnosis and teleconsultation, were mentioned in seven studies, utilizing images and information submission services to platforms, WhatsApp or applications. One study presented teleconsultations involving slides and another study introduced teleconsultation mediated by a professional. Digital strategies telediagnosis and teleconsultations enable the histopathological diagnosis of oral cancer through clinical or histopathological images. The higher the observed diagnostic agreement, the better the performance of the strategy.

## INTRODUCTION

The availability of Information and Communication Technologies (ICTs) and their overall accessibility have made the implementation of remote health services possible. This provides the conditions for responding to the contemporary challenges of universal

Corresponding author
Rebeka Thiara Nascimento dos Santos, rebeka.santos@upe.br

healthcare systems, such as the overload on specialized services, the high volume of referrals, long waiting periods and the need for travel, combined with the shortage of specialized professionals. Telediagnosis (TD) and teleconsultation (TC) services help front-line professionals make decisions, manage patients and provide them with comprehensive care (*Ghai, 2020*).

The TD utilizes ICTs to provide diagnostic support services across geographical and temporal distances, facilitating the exchange of files and images (*Ghai, 2020*; *Ali & El Ansari, 2022*). The TC is a registered consultation conducted among healthcare workers, professionals, patients and managers in the healthcare field, utilizing bidirectional telecommunication instruments. It aims to clarify doubts regarding clinical procedures, healthcare actions, and work-related issues. The TC can occur both in real-time and off-line *via* messaging or emails (*Ghai, 2020*; *Caldarelli & Haddad, 2016*; *Paixão et al., 2018*). In recent years, studies have been conducted to evaluate the potential of telehealth in the early diagnosis of potentially malignant or malignant oral lesions (OL), yielding promising results (*Flores et al., 2022*; *Uhrin et al., 2023*).

Oral squamous cell carcinoma (OSCC), squamous cell carcinoma (SCC) of the mouth or oral cancer (OC) represents 90 to 95% of cases of malignant oral neoplasms (*Motta Rda et al., 2009*; *Chamoli et al., 2021*). It is responsible for 30% of cancer-related deaths in low and middle-income countries (*Warnakulasuriya, 2009*). Initially, the oral cavity is examined for observation of suspicious lesions. However, for the diagnosis of OSCC, histopathological analysis from a biopsy is necessary (*Chamoli et al., 2021*).

Factors such as variable presentation patterns of lesions, professional uncertainty, and difficult access to specialized services, lead to delayed diagnosis resulting in an unfavorable long term prognosis (*Ergun et al., 2009*; *Warnakulasuriya et al., 1984*). The objective of this study was to conduct a scoping review and include different research designs to map the existing literature on digital auxiliary strategies used to assist in the diagnosis of OSCC.

## METHODS

The structure of this scoping review was on the five methodological steps proposed by *Arksey & O'Malley (2005)*: identification of the research question, identification of relevant studies, selection of studies, mapping and grouping of data, summarizing and reporting results. Additionally, it followed the Joanna Briggs Institute Manual for Evidence Synthesis and the Preferred Reporting Items for Systematic Reviews and Meta-Analyses extension for Scoping Reviews checklist (PRISMA-ScR) (*Tricco et al., 2018*). The methods were registered on the Open Science Framework (osf.io/qweu8).

The research question was: "What digital strategies have been used to assist in the diagnosis of oral squamous cell carcinoma?" The search strategy was not restricted to a specific publication period. Electronic searches were independently conducted by two authors (R.T.N.S. and C.A.B.F.) across the following databases: PubMed/MEDLINE, Scopus, Web of Science, Embase, and ScienceDirect. All searches were conducted on 18 June 2023. The search strategies used to electronically search are shown in Table 1. No filters were applied.

**Table 1  Search strategies and date used to electronically search for articles.**

| Database | Search strategies |
|---|---|
| **Pubmed/Medline** | #1 (Diagnosis OR Diagnosis, Oral OR Diagnoses OR Diagnose OR Early Diagnosis OR Diagnosis, Early OR Early Detection OR Cancer Early Detection OR Cancer Screening OR Screening, Cancer) |
| | #2 (oral cavity cancer OR oral cavity neoplasm OR oral cavity tumor OR oral cavity tumour OR mouth neoplasms OR tongue cancer OR tongue neoplasm OR tongue tumour OR tongue tumor OR mouth neoplasms OR Carcinoma, Squamous Cell OR Carcinomas, Squamous Cell OR Squamous Cell Carcinomas OR Squamous Cell Carcinoma OR Epidermoid Carcinoma OR Epidermoid Carcinomas OR Carcinoma, Planocellular OR Carcinomas, Planocellular OR Planocellular Carcinoma OR Planocellular Carcinomas) |
| | #3 (Teledentistry OR mhealth OR ehealth OR ehealth, technology OR Telemedicine OR Telehealth OR Digital Health OR Teleassistance OR remote consultation OR Teleconsultation OR Teleconsultations OR Telediagnosis OR Telepathology OR Telementoring OR Teledental Care OR Telemonitoring); |
| | #1 AND #2 AND #3. |
| **Scopus** | ''Diagnosis, Oral'' OR ''Early Detection of cancer'' AND ''Mouth Neoplasms'' OR ''Tongue Neoplasms'' OR ''Carcinoma, Squamous Cell'' AND Telemedicine OR ''Remote Consultation'' OR Telepathology |
| **Web of Science** | 'Diagnosis' OR 'Diagnosis, Oral OR Diagnoses' OR 'Diagnose' OR 'Early Diagnosis' OR 'Diagnosis, Early' OR 'Early Detection' OR 'Cancer Early Detection' OR 'Cancer Screening' OR 'Screening, Cancer') AND ('oral cavity cancer' OR 'oral cavity neoplasm' OR 'oral cavity tumor' OR 'oral cavity tumour' OR 'mouth neoplasms' OR 'tongue cancer' OR 'tongue neoplasm' OR 'tongue tumour' OR 'tongue tumor' OR 'mouth neoplasms' OR 'Carcinoma, Squamous Cell' OR 'Carcinomas, Squamous Cell' OR 'Squamous Cell Carcinomas' OR 'Squamous Cell Carcinoma' OR 'Epidermoid Carcinoma' OR 'Epidermoid Carcinomas' OR 'Carcinoma, Planocellular' OR 'Carcinomas, Planocellular' OR 'Planocellular Carcinoma' OR 'Planocellular Carcinomas') AND ('Teledentistry' OR 'mhealth' OR 'ehealth' OR 'health, technology' OR 'Telemedicine' OR 'Telehealth' OR 'Digital Health' OR 'Teleassistance' OR 'remote consultation' OR 'Teleconsultation' OR 'Teleconsultations' OR 'Telediagnosis' OR 'Telepathology' OR 'Telementoring' OR 'Teledental Care' OR 'Telemonitoring' |

**Table 1** (*continued*)

| Database | Search strategies |
|---|---|
| **Embase** | Diagnosis OR 'diagnosis related group' OR 'early diagnosis' OR 'early cancer diagnosis' OR 'cancer screening' AND 'mouth cancer' OR 'mouth disease' OR 'mouth tumor' OR 'tongue cancer' OR 'squamous cell carcinoma' AND teledentistry OR telehealth OR telemedicine OR teleconsultation OR telediagnosis OR telepathology OR telemonitoring |
| **ScienceDirect** | "diagnosis oral" OR "early detection of cancer" AND "Mouth Neoplasms" OR "Tongue Neoplasms" OR "Carcinoma, Squamous Cell" AND Telemedicine OR Remote Consultation OR Telepathology |

The same authors conducted a manual search in Telemedicine, Oral Pathology and Public Health journals, including PeerJ, The Journal of the American Dental Association, Telemedicine and e-Health, Oral Diseases, Scientific Reports, British Journal of Cancer, International Journal of Cancer and Frontiers in Public Health.

Inclusion criteria comprised of articles on digital strategies, specifically TD, TC—synchronous or asynchronous—or Teleconsultation Mediated by a Professional (TCMP) in the diagnosis of OSCC. For this study, the Population, Concept, and Context (PCC) framework was utilized: Population—patients with OSCC; Concept—Diagnosis; Context—digital strategies. Exclusion criteria included review articles, letters, conference abstracts and editorials, as well as studies addressing direct TC between professionals and patients, telemanagement and teleducation.

The study selection followed the Preferred Reporting Items for Systematic Reviews and Meta-Analyses extension for Scoping Reviews (PRISMA-ScR) (*Tricco et al., 2018*): two authors (R.T.N.S. and C.A.B.F.) independently reviewed titles and abstracts of all identified references, applying eligibility criteria (blinded process). Disagreements were resolved through discussions with a third author (A.P.V.S.). The Rayyan software was employed.

The variables collected from the studies included: author(s), year of publication, country, study design, service nomenclature, strategy, tools, validation method in the detection of OSCC and whether agreement between assessments was performed. After mapping the relevant information from the studies, a basic numerical analysis was conducted to assess the extent, nature, and distribution of the studies included in the review (flowchart and infographic figures). Individualized result tables were formulated for the included studies, highlighting data related to their characteristics. The results were categorized into themes corresponding to the study findings.

# RESULTS

The electronic search in June 2023 yielded 1,798 articles across the consulted databases, as depicted in the flowchart (Fig. 1). In total, 16 studies were included in this review: five cross-sectional, five retrospective cross-sectional, two prospective cross-sectional, one screening intervention and three retrospective cohorts. The compiled results take
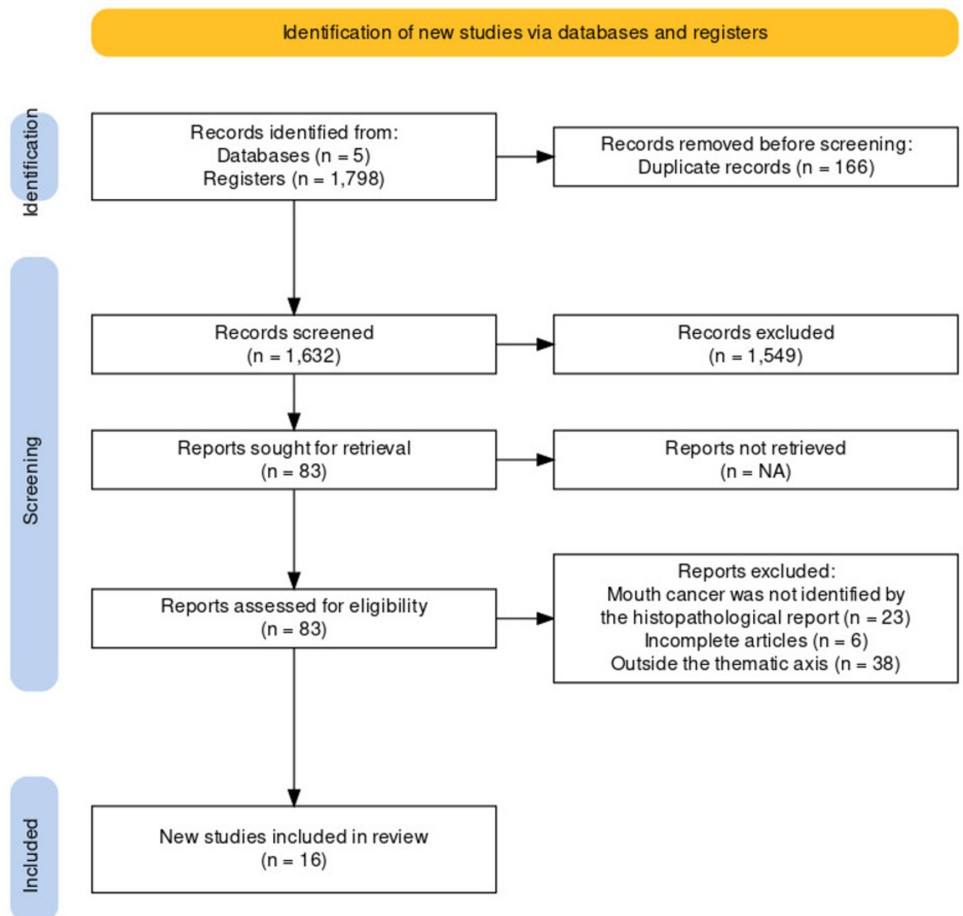

**Figure 1** Screening and enrollment PRISMA-ScR flow diagram showing selection of articles for scoping review.

into account that all selected studies involved remote identification and histopathological examination of OSCC. These are qualitatively summarized in Table 2 and quantitatively presented in Table 3.

TD was reported in nine studies (*Bhatt et al., 2018*; *Czerninski, Netanel & Basile, 2022*; *Torres-Pereira et al., 2013*; *Petruzzi & De Benedittis, 2016*; *Birur et al., 2022*; *Tanveer & Garg, 2021*; *Garg, Tanveer & Gaur, 2019*; *Skandarajah et al., 2017*; *Sunny et al., 2019*). The tools employed included data screening in prototype applications, such as prototypes for SIM called Mobile Health or mHealth (*Bhatt et al., 2018*), clinical images captured by digital cameras (*Czerninski, Netanel & Basile, 2022*; *Torres-Pereira et al., 2013*), mobile phones (*Petruzzi & De Benedittis, 2016*) or Artificial Intelligence (AI), through cellphones and probe integrated with mobile and cloud platforms (*Birur et al., 2022*). The exchange of histopathological images *via* ICTs was also featured in four studies (*Tanveer & Garg, 2021*; *Garg, Tanveer & Gaur, 2019*; *Skandarajah et al., 2017*; *Sunny et al., 2019*).

TD associated with TC was reported in seven studies, involving services for sending images and information to platforms, such as cloud-based platform by e-mail (*Flores et*
**Table 2 Characteristics of the included studies on digital strategies to assist in the diagnosis of OSCC.**

| Authors | Country | Study design | Service Nomenclature | Strategy | Tools | Validation in the detection of OSCC | Rated Concordance? |
|---------|---------|--------------|----------------------|----------|-------|-------------------------------------|--------------------|
| *Czerninski, Netanel & Basile (2022)* | Israel | Cross-sectional | Clinical image evaluation. | TD through on-screen image assessment. | Oral digital images were captured using a Canon 1200D digital camera (macro lens 105/2.8, aperture F18, shutter speed 1/80), with a ring flash. These images were displayed on a screen and assessed through an electronic questionnaire. | Biopsy with histopathological diagnosis. | No. |
| *Flores et al. (2022)* | Brazil | Prospective cross-sectional | EstomatoNet Program. | TD through a report with diagnostic hypothesis and ATC through recommendations. | Oral images were captured using a mobile phone and sent to a cloud-based platform via e-mail. | Biopsy with histopathological diagnosis. | Yes. Agreement between in-person consultation and tele-diagnosis, both conducted by specialists. |
| *Birur et al. (2022)* | India | Prospective cross-sectional | MobileNet | TD through image screening based on artificial intelligence. | Device with dual-modality artificial neural network imaging, simple (cell phone) and complex (probe), integrated with both mobile and cloud platforms. | Incisional biopsy and histopathological evaluation. | Yes. Agreement between on-site examination and tele-diagnosis, both conducted by a professional. |
| *Tanveer & Garg (2021)* | India | Retrospective cross-sectional | Smartphone-assisted telepathology. | TD through pathological assessment of slide images. | Slides from a microscope eyepiece were captured with a mobile phone, transferred and archived via Google Drive. They were then assessed on a 40-inch LED television screen connected to a laptop via an HDMI cable. | Biopsy with histopathological diagnosis. | Yes. Intra-observer agreement between microscopic diagnosis and imaging diagnosis, by the same specialist. |
| *Araújo et al. (2021)* | Brazil | Retrospective cross-sectional | Digital Pathology. | TD through remote reporting, and pathological STC through online meeting. | Slide digitization and discussion via Google Meet. | Google Meet and discussion of digitized slide cases. | Yes. Intra-observer agreement between the DM and CM. |

**Table 2** (*continued*)

| Authors | Country | Study design | Service Nomen-clature | Strategy | Tools | Validation in the detection of OSCC | Rated Concor-dance? |
|---------|---------|--------------|------------------------|----------|-------|-------------------------------------|---------------------|
| *Fonseca et al. (2021)* | Brazil | Cross-sectional | Telediagnosis of oral lesions. | TD through hypotheses and ATC through recommendations. | OL images were captured with a mobile phone and archived on a cloud platform - Dropbox. | Incisional or excisional biopsy with histopathological diagnosis. | Yes. Agreement between in-person consultation and telediagnosis, both by specialists. |
| *Perdoncini et al. (2021)* | Brazil | Cross-sectional | Synchronous teleconsultation in oral medicine. | Synchronous TCMP | OL images captured by a mobile phone and sent via WhatsApp. | Biopsy with histopathological diagnosis. | Yes. Agreement between Telediagnosis and in-person diagnosis, both by specialists. |
| *Tamba et al. (2021)* | Senegal | Cross-sectional | "WhatsApp Dentists of Senegal" | TD through identification of the shared condition and TC (both ATC and STC) through recommendation of the course of action. | OL images captured by a mobile phone and sent via WhatsApp. | Biopsy with histopathological diagnosis. | No. |
| *Sunny et al. (2019)* | India | Retrospective cohort | Telecytology with Convolutional Neural Network (CNN) "CellScope". | Cytological TD through image pre-processing algorithm. | Digitalized cytological slide images using a tablet-iPad from a mobile microscope, sent via a web server, combined with a Convolutional Neural Network (CNN) for atypical cell classification and a risk stratification model based on Artificial Neural Network (ANN). | Cytology (direct microscopy) and histopathological diagnosis. | Yes. Agreement between integrated telecytology with risk stratification model, direct microscopy, and neoplastic histology. |
| *Garg, Tanveer & Gaur (2019)* | India | Retrospective cross-sectional | Smartphone-assisted telepathology. | Histopathological TD through visualization of slide images. | Slide images from the eyepiece of a microscope captured by a mobile phone and sent via WhatsApp. | Conventional microscopy for viewing on glass slides. | Yes. Agreement between conventional microscopy performed by viewing on glass slides and WhatsApp-based diagnosis. |
| *Carrard et al. (2018)* | Brazil | Cross-sectional | EstomatoNet Program | TD through a report with diagnostic hypothesis and ATC through recommendations. | Oral images captured by a mobile phone and sent to a cloud-based platform via email. | Partial biopsy with histopathological diagnosis. | Yes. Agreement between the opinion of the requesting professional through clinical examination and specialized teleconsultants through remote diagnosis. |

**Table 2** (*continued*)

| Authors | Country | Study design | Service Nomenclature | Strategy | Tools | Validation in the detection of OSCC | Rated Concordance? |
|---|---|---|---|---|---|---|---|
| *Bhatt et al. (2018)* | India | Screening interventional | Mobile Health or 'mHealth' | TD through screening. | Data collection forms in a prototype SIM card mobile application. | Biopsy with histopathological diagnosis. | No. |
| *Skandarajah et al. (2017)* | India | Retrospective cohort | Tablet-based Tele-cytology "CellScope" | Pathological TD through review of slide images. | Digitized cytological brush biopsy slides using a mobile microscope based on a tablet with iPad Mini, combined and sent via a web server. | Conventional microscopy cytology and histology. | Yes. Agreement between gold standard histology, conventional microscopy cytology, and telepathology review. |
| *Petruzzi & De Benedittis (2016)* | Italy | Retrospective cross-sectional | WhatsApp for oral pathology and medical centers. | TD through image assessment. | Clinical images captured on a mobile phone and sent via WhatsApp. | Biopsy with histopathological diagnosis. | Yes. Agreement between telemedicine impression and clinical-pathological diagnosis. |
| *Birur et al. (2015)* | India | Retrospective cohort | Oncogrid - Remote Oral Cancer Surveillance Program based on mobile phone. | TD through screening and ATC through guidance in short text messages. | Intraoral lesions and data recorded via mobile phone, uploaded to the mobile platform (Sana). | Biopsy with histopathological diagnosis. | No. |
| *Torres-Pereira et al. (2013)* | Brazil | Retrospective cross-sectional | Telediagnosis in oral medicine. | TD through the evaluation of clinical images with hypothesis generation. | Oral digital photographs taken with a digital camera (EOS 300 Rebel; Canon, Tokyo, Japan) equipped with a 100 mm macro lens and a Canon circular flash system; then sent via email to teleconsultants. | Biopsy with histopathological diagnosis. | Yes. Agreement between expert consultants and histopathological diagnosis. |

**Notes.**

TD, Telediagnosis; TC, Teleconsultation; STC, Synchronous Teleconsultation; ATC, Asynchronous Teleconsultation; TCMP, Teleconsultation Mediated by a Professional; DM, Digital Method; CM, Conventional Method.
**Table 3  Quantitative results of the of the included studies on digital strategies to assist in the diagnosis of OSCC.**

| Authors | Country | Study design | Strategy | N° of total cases | N° of OSCC cases | Agreement ($\kappa$) | Agreement classification |
|---|---|---|---|---|---|---|---|
| Czerninski, Netanel & Basile (2022) | Israel | Cross-sectional | TD through on-screen image assessment. | 18 | 12 | N/A | N/A |
| Flores et al. (2022) | Brazil | Prospective cross-sectional | TD through a report with diagnostic hypothesis and ATC through recommendations. | 100 | 10 | 0.95 | Almost perfect |
| Birur et al. (2022) | India | Prospective cross-sectional | TD through image screening based on artificial intelligence. | 752 | 61 | 0.79 | Substantial |
| Tanveer & Garg (2021) | India | Retrospective cross-sectional | TD through pathological assessment of slide images. | 90 | 62 | N/A | N/A |
| Araújo et al. (2021) | Brazil | Retrospective cross-sectional | TD through remote reporting, and pathological STC through online meeting. | 162 | 12 | 0.85–0.98 | Almost perfect |
| Fonseca et al. (2021) | Brazil | Cross-sectional | TD through hypotheses and ATC through recommendations. | 113 | 1 | 0.817–0.903 | Almost perfect |
| Perdoncini et al. (2021) | Brazil | Cross-sectional | Synchronous TCMP | 41 | 1 | 0.922 | Almost perfect |
| Tamba et al. (2021) | Senegal | Cross-sectional | TD through identification of the shared condition and TC (both ATC and STC) through recommendation of the course of action. | 150 | 53 | N/A | N/A |
| Sunny et al. (2019) | India | Retrospective cohort | Cytological TD through image preprocessing algorithm. | 2765 | 20 | 0.67–0.72 | Substantial |
| Garg, Tanveer & Gaur (2019) | India | Retrospective cross-sectional | Histopathological TD through visualization of slide images. | 100 | 58 | N/A | N/A |
| Carrard et al. (2018) | Brazil | Cross-sectional | TD through a report with diagnostic hypothesis and ATC through recommendations. | 259 | 22 | N/A | N/A |
| Bhatt et al. (2018) | India | Screening interventional | TD through screening. | 8516 | 490 | N/A | N/A |
| Skandarajah et al. (2017) | India | Retrospective cohort | Pathological TD through review of slide images. | 32 | 19 | 0.695 | Substantial |

| Authors | Country | Study design | Strategy | N° of total cases | N° of OSCC cases | Agreement ($\kappa$) | Agreement classification |
|---|---|---|---|---|---|---|---|
| *Petruzzi & De Benedittis (2016)* | Italy | Retrospective cross-sectional | TD through image assessment. | 96 | 7 | N/A | N/A |
| *Birur et al. (2015)* | India | Retrospective cohort | TD through screening and ATC through guidance in short text messages. | 236 | 6 | N/A | N/A |
| *Torres-Pereira et al. (2013)* | Brazil | Retrospective cross-sectional | TD through the evaluation of clinical images with hypothesis generation. | 60 | 5 | 0.669/0.574 | Substantial/Moderate |

**Notes.**
OSCC, Oral Squamous Cell Carcinoma; TD, Telediagnosis; TC, Teleconsultation; STC, Synchronous Teleconsultation; ATC, Asynchronous Teleconsultation; TCMP, Teleconsultation Mediated by a Professional; $\kappa$, kappa.

*al., 2022*; *Carrard et al., 2018*) and cloud platform—Dropbox (*Fonseca et al., 2021*) cross-platform instant messaging called WhatsApp (*Tamba et al., 2021*) or applications, such as Remote Oral Cancer Surveillance Program based on mobile phone (*Birur et al., 2015*). One study presented TC involving slides (*Araújo et al., 2021*), and another study introduced professional-mediated TC (*Perdoncini et al., 2021*).

Regarding the agreement observed in the histopathological diagnosis, optimal performance was found when images captured by mobile phones (*Flores et al., 2022*; *Fonseca et al., 2021*) or digitized slides (*Araújo et al., 2021*) were sent to a platform or by e-mail for specialists assessment.

Figure 2 presents an infographic representing the strategies with their respective nomenclatures. Figure 3 illustrates the tools for the diagnosis of OSCC. Figure 4 maps the countries most interested in the topic.

## Telediagnosis
### Data screening
A prototype for SIM, configured to support data collection forms carried out by community health workers (CHWs), tracked 8,516 patients. Of these, 5% ($n = 490$) underwent biopsies and were diagnosed with OSCC (*Bhatt et al., 2018*).

### Clinical images
Two studies (*Czerninski, Netanel & Basile, 2022*; *Torres-Pereira et al., 2013*) presented images of OL captured by a digital camera. The images were sent *via* e-mail to oral medicine specialists (*Czerninski, Netanel & Basile, 2022*; *Torres-Pereira et al., 2013*) and dental students (*Czerninski, Netanel & Basile, 2022*). The results of the assessment of malignant images of OSCC showed, on average, only 1.2% ($\pm$ SD1.3) cancer images were correctly recognized (*Czerninski, Netanel & Basile, 2022*). Meanwhile, *Torres-Pereira et al. (2013)* sent case images to two different teleconsultants, who recorded two diagnostic hypotheses. All of them were correct for OSCC.

Images were also captured on cell phones by dental professionals and sent *via* WhatsApp to two teleconsultants. All cases with suspected carcinoma were confirmed

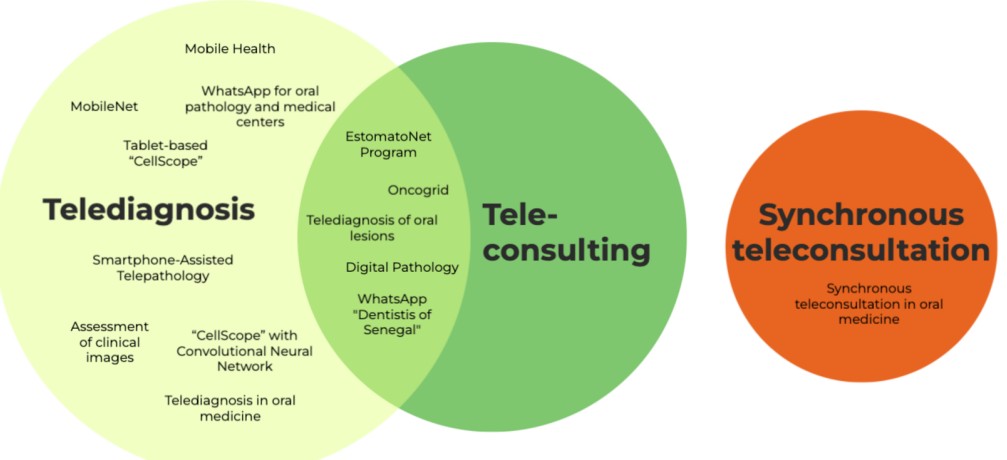

**Figure 2** **Infographic figure of the nomenclatures associated with auxiliary digital strategies used to assist in the diagnosis of OSCC.**

after biopsy (*Petruzzi & De Benedittis, 2016*). Similarly, AI assessed images through an artificial neural network device, integrated with mobile phones and cloud for evaluation by remote specialists. The results showed that out of 752 assessed by on-site experts, 61 were diagnosed as OSCC or carcinoma *in situ*. Remote experts demonstrated, when compared to the local expert, a concordance of 0.79 (CI [0.75–0.844]) for OSCC (*Birur et al., 2022*). Regarding the concordance between the remote clinical impression and the final histological diagnosis, the related statistics were presented in two articles, summarized in Table 3 (*Torres-Pereira et al., 2013*; *Fonseca et al., 2021*).

### Histopathological images

Four studies reported the use of digitized histopathological images (*Tanveer & Garg, 2021*; *Garg, Tanveer & Gaur, 2019*; *Skandarajah et al., 2017*; *Sunny et al., 2019*). Two presented digitized slides through mobile phones: *Tanveer & Garg (2021)* photographed 62 slides of OSCC from a microscope eyepiece and sent them *via* Google Drive. The images were transferred to the same pathologist. All cases of OSCC were correctly diagnosed (62/62). (*Garg, Tanveer & Gaur, 2019*) photographed 58 slides of OSCC and sent the images *via* WhatsApp to a pathologist who was unaware of the cases, and he responded by sending the diagnosis. The concordance rate for OSCC was 96.6% (56/58).

*Skandarajah et al. (2017)* and *Sunny et al. (2019)* also digitized cytological slides using a tablet-iPad, a mobile microscope called "CellScope", and sent them *via* a web server (*Skandarajah et al., 2017*). *Skandarajah et al. (2017)* evaluated a total of 19 images of OSCC, with substantial agreement of 0.695.

In addition to using the mobile microscope, *Sunny et al. (2019)* also performed a combination with the Convolutional Neural Network (CNN) and the model of risk stratification based on Artificial Neural Network (ANN). In this case, in 38 neoplastic

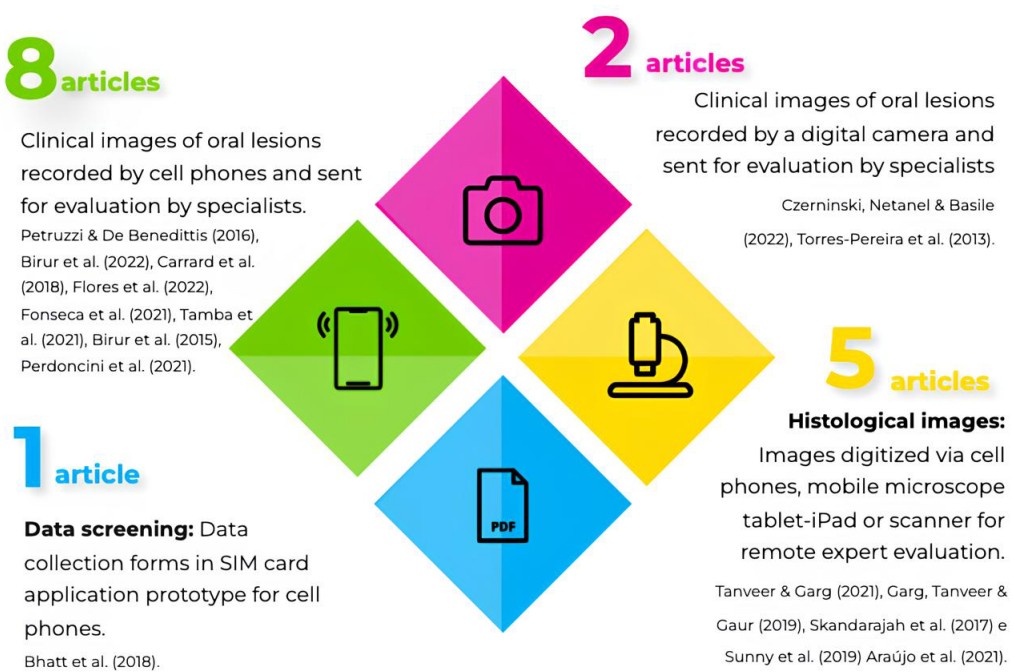

**Figure 3** **Infographic figure of digital tools in OSCC diagnosis.** *Petruzzi & De Benedittis, 2016*; *Birur et al., 2022*; *Carrard et al., 2018*; *Flores et al., 2022*; *Fonseca et al., 2021*; *Tamba et al., 2021*; *Birur et al., 2015*; *Perdoncini et al., 2021*; *Bhatt et al., 2018*; *Czerninski, Netanel & Basile, 2022*; *Torres-Pereira et al., 2013*; *Tanveer & Garg, 2021*; *Garg, Tanveer & Gaur, 2019*; *Skandarajah et al., 2017*; *Sunny et al., 2019*; *Araújo et al., 2021*.

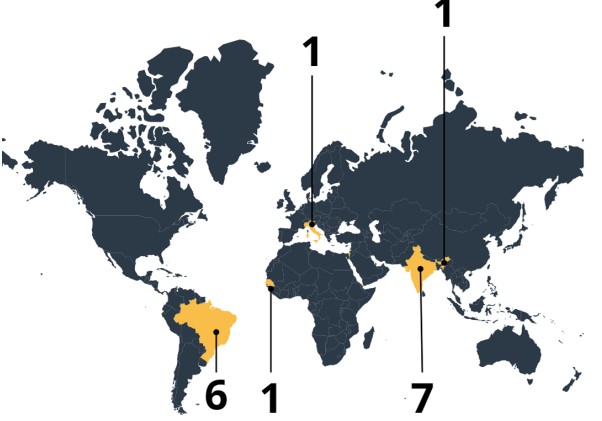

**WHO IS RESEARCHING DIGITAL STRATEGIES?**

Check out some nations that stand out in scientific production:

- India: 7
- Brazil: 6
- Senegal: 1
- Italy: 1
- Israel: 1

**Figure 4** **Infographic figure representing the countries most interested in the topic.** Map created in Canva.

cases, 92% ($n = 35$) were diagnosed by both pathologists through direct microscopy and telecytology and showed good sensitivity and specificity when compared to histopathology, demonstrating an substantial agreement of 0.67–0.72.

### Telediagnosis associated with teleconsultation
#### Clinical images
The TD associated with TC, was reported in five studies (*Carrard et al., 2018*; *Fonseca et al., 2021*; *Tamba et al., 2021*; *Birur et al., 2015*; *Araújo et al., 2021*). Two of them presented the use of the *EstomatoNet* service. The clinical information and photographs of oral lesions were sent through a cloud-based platform, and specialized teleconsultants received the data, provided a diagnostic hypothesis, and recommended a course of action. *Carrard et al. (2018)* diagnosed OSCC in 8.5% of cases ($n = 22$), and *Flores et al. (2022)* in 10% of cases ($n = 10$).

The agreement between face-to-face consultations and TD was 72.2% and 95%, respectively. In the study by *Fonseca et al. (2021)* remote analysis of OL was carried out using a Dropbox folder containing clinical information and photographs of the lesions. These were sent to three assessors for one or two diagnostic hypotheses for each case. The assessors also sent referral decisions, requests for additional tests, diagnostic difficulties and image quality. For the OSCC case ($n = 1$), the diagnostic hypothesis was similar for all the assessors, whereas for the other OL's, the agreement was almost perfect (0.817–0.903).

*Tamba et al. (2021)* conducted communications between healthcare and dental professionals and specialists through WhatsApp. Data was collected using data collection forms. The means for these information exchanges included text, radiographs, photographs, audio (voice messages), and video. One hundred and fifty communications were included, with 35% of the tumors being malignant, including OSCC.

*Birur et al. (2015)* employed a mobile health (mHealth) system based on a mobile App to input and store patient demographic data, risk factors, and symptoms. This screening was conducted by CHWs and dentists. Out of 129 patients referred for biopsy, 6 had malignant lesions, showing 100% agreement with the specialist.

Regarding TCMP, this was reported in one study (*Perdoncini et al., 2021*), involving patients with OL referred for specialized care. They were initially assessed in person by a general practitioner who obtained photographs of the lesions with a smartphone. The images were sent through a mobile App to an oral medicine specialist, followed by a video call. After interviewing the patient, the specialist formulated a diagnostic hypothesis and provided suggestions for case management. A second specialist, blinded to the initial assessment, personally evaluated the OL and defined a diagnosis. The hypothesis for the case of OSCC ($n = 1$) was the same for both TC and the clinical examination among the evaluators.

### Histopathological images
In addition to TD, *Araújo et al. (2021)* introduced remote reporting of slides associated with TC through an online meeting *via* Google Meet. This involved the discussion of cases to validate the remote diagnosis. The intra-observer agreement between the digital method (DM) and the conventional method (CM) was considered nearly perfect (0.85–0.98).

## DISCUSSION

Despite emerging scientific evidence, there is still a gap in the knowledge of digital communication strategies between professionals about OLs. These barriers become potential within the theme of OSCC. It is crucial to understand these services and the facilitators in this diagnostic process. This review mapped the literature and included studies on OSCC diagnosis and the use of ICTs. The findings included both cases of clinical images that led to a histopathological report and slide scanning that confirmed the diagnosis. The scope of the study highlights its clinical relevance.

Digital strategies in dentistry have proved very useful. Most of the time, OLs are evident and can be captured in form of photographic documentation (*Maret et al., 2020*). In a multicenter study conducted by *Flores et al. (2022)*, the agreement among examiners was $\kappa = 0.95$, indicating almost perfect concordance. These results show the excellent performance of TD associated with TC (*Flores et al., 2022*; *Fonseca et al., 2021*). Best results are achieved when the oral images captured by a mobile phone are sent to a cloud-based platform *via* email, making remote support a promising alternative to in-person consultations (*Fonseca et al., 2021*).

Similarly, the Brazilian study by *Tamba et al. (2021)* evaluated the reliability between requesting dentists and specialists in TD through image analysis provided by specialized teleconsultants on a cloud-based platform called TelessaúdeRS. The agreement between the requester and the teleconsultant's opinion was 72.2% (96 cases out of the total 131 cases). Their findings supported the feasibility of using telemedicine in primary oral health care to assist in decision-making at the primary care level.

Current evidence (*Conceição et al., 2010*; *Araújo et al., 2018*; *Goacher et al., 2017*) indicates that images of whole slides are suitable for histopathological diagnosis with performance comparable to light microscopy in various subspecialties of human pathology, including head and neck pathology. The present study evinces that TD is a valuable tool in the interpretation of histopathological slides. *Araújo et al. (2021)*, through a cross-sectional study, sought to validate the remote evaluation of whole histopathological slides and obtained a $\kappa$ ranging from 0.85 to 0.98, with a 95% confidence interval ranging from 0.81 to 1.

*Garg, Tanveer & Gaur (2019)*, in their study conducted in 2019, aimed to assess the utility of transferring WhatsApp images in the histopathological diagnosis of common OL. They observed that a significant limitation of this tool is the reduction in file size and, consequently, image quality. While WhatsApp offers the advantage of transferring a large number of files in a short time, the authors emphasize that prioritizing faster transmission at the expense of lower image quality has a negative impact on case interpretation. The authors suggest Google Drive as an alternative for sharing images for interpretation.

Corroborating the findings of *Garg, Tanveer & Gaur (2019)*, a study conducted by *Tanveer & Garg (2021)* aimed to evaluate the intra-observer agreement of telepathological diagnosis of OL using a new configuration associated with conventional microscopy diagnosis. The study found that Google Drive is a valuable tool for image transfer. The authors highlight its advantages, such as sharing images with large dimensions,

high quality, and the ability to view and enlarge them on high-resolution computers and televisions. Similarly, *Skandarajah et al. (2017)* observed that cloud storage of images provides file quality maintenance, sharing among evaluators, and remote analysis of cases by pathologists.

The studies on histopathological images included in the sample of the study (*Tanveer & Garg, 2021*; *Garg, Tanveer & Gaur, 2019*; *Skandarajah et al., 2017*; *Sunny et al., 2019*; *Araújo et al., 2021*) showed consistent results in TD and TC. They indicate that digital pathology can also be a useful tool for remote discussion of cases, consultation and opinion of other specialists, planning patient management and sharing with students. Thus, it represents an alternative and reliable diagnostic method (*Araújo et al., 2021*).

Two studies (*Czerninski, Netanel & Basile, 2022*; *Torres-Pereira et al., 2013*) presented TD based on images of OL captured by a digital camera. The images were sent *via* e-mail through a link to oral medicine specialists (*Czerninski, Netanel & Basile, 2022*; *Torres-Pereira et al., 2013*) and dental students (*Czerninski, Netanel & Basile, 2022*) for evaluation and response to a questionnaire about the diagnosis. The results of the assessment of the images of OL were better among clinicians when compared to the students. In order to verify the difference between the evaluation of oral medicine specialists and dental students, we highlight the question of why dental students, who are not specialists in oral medicine or oral diagnosis, were included in the study to provide an opinion on the diagnosis of OL. The authors infer they aimed to enable the analysis of the effects of other factors, such as experience and interest in oral pathology, in the evaluation (*Czerninski, Netanel & Basile, 2022*). Still, the results highlight that the higher rate of diagnostic accuracy is a reflection of the professional exercise of the specialists (*Czerninski, Netanel & Basile, 2022*).

Regarding the importance of the visual resource used for evaluation, the irregularity of the lesion resulted in a correct diagnosis, while incorrect responses were related to changes in color. This demonstrates that the use of clinical images as part of the diagnostic process provides good results, although it may be necessary to increase clinical experience for both graduates and undergraduates to improve diagnostic accuracy (*Czerninski, Netanel & Basile, 2022*).

In addition to the aforementioned, one study aimed to assess the clinical utility/effectiveness of a telecytology system combined with a risk stratification model based on ANN as an alternative TD method, aiming for the early detection of malignant lesions (*Sunny et al., 2019*). With the use of ANN, it was observed that each cell took approximately one second to be classified, and on average, 3 min to categorize all cells from a patient, showing an accuracy of 90% in delineating OSCC. Therefore, this method increased the overall system accuracy by 30% when compared to the in-person method (*Sunny et al., 2019*).

The diagnosis of OC through digital strategies and tools is an innovative and essential approach that enables early diagnosis and effective treatment of this disease. The publication of studies in this area has been a priority in various countries. India has emerged as a leader in our results. Brazil also stands out in this scenario, demonstrating an interest in applying these strategies to improve access to early diagnosis of OSCC, which is particularly relevant due to its vast territory and accessibility challenges. Although with a smaller number of

publications from Senegal, Italy, and Israel, it is understood that these countries recognize the importance of the topic and are researching its benefits and applications.

Despite the very promising evidence, it is crucial to consider the different socio-economic realities of the regions where these strategies are possible to be implemented. From the perspective of the Asian continent, *Anand et al. (2022)* points out that a significant portion of the population suffers from oral submucous fibrosis—a disorder with malignant transformation potential—that can manifest as OSCC concurrently. However, the lack of access to technologies, coupled with poor or nonexistent training for healthcare professionals regarding photographic registration protocols, may lead to underdiagnosis. These limitations must be taken into account. Therefore, the planning of the implementation of these services should be meticulous.

An important factor to consider in the studies evaluated is that there is no mention or discussion over the legal and ethical aspects of identity privacy and their personal data in the teleconsultations. The majority of the studies included were conducted in public hospitals and universities. Thus, considering their role and contribution to public health systems, it is suggested that these services have their own terms and conditions to this end. The confidentiality of the registered data of the services analyzed is necessary to safeguard the privacy of the patients. Due to the potential risk regarding licensing, jurisdiction, and bad practice, the patients should be properly clarified, being conscious of the inherent risk of inadequate diagnosis or treatment due to technological limitations (*Deshpande et al., 2021*).

Authors need to assume that the search was carried out by authors at different postgraduate levels—Master's and PhD students—as limitations of the review process used. This may cause divergences in the selection of studies even if a third reviewer has evaluated it.

## CONCLUSIONS

Both TD and TC are emerging as alternative and complementary means in the diagnosis and decision-making process regarding OSCC, showing agreement with existing gold standard techniques. The optimal performance of these strategies occurs when oral images captured by cellphones or digitized slides are sent and discussed by experts through a cloud-based platform. The TD proves to be a valuable option for remote pathological reports. The TC has a significant application in supporting frontline professionals in the presence of clinical OL.

### Funding
The authors received no funding for this work.

### Competing Interests
The authors declare there are no competing interests.

## Author Contributions

- Rebeka Thiara Nascimento dos Santos conceived and designed the experiments, performed the experiments, analyzed the data, prepared figures and/or tables, authored or reviewed drafts of the article, and approved the final draft.
- Caroline Augusta Belo Faria conceived and designed the experiments, analyzed the data, authored or reviewed drafts of the article, and approved the final draft.
- Nathalya Fedechen Martins performed the experiments, prepared figures and/or tables, authored or reviewed drafts of the article, and approved the final draft.
- Luiz Gustavo de Sousa Duda Júnior performed the experiments, prepared figures and/or tables, authored or reviewed drafts of the article, and approved the final draft.
- Ana Beatriz Fernandes Azevêdo performed the experiments, authored or reviewed drafts of the article, and approved the final draft.
- Weslay Rodrigues da Silva conceived and designed the experiments, analyzed the data, authored or reviewed drafts of the article, and approved the final draft.
- Ana Paula Veras Sobral conceived and designed the experiments, analyzed the data, authored or reviewed drafts of the article, and approved the final draft.

## Data Availability

This is a literature review.

## Supplemental Information

Supplemental information for this article can be found online at http://dx.doi.org/10.7717/peerj.17329#supplemental-information.

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
