# Peer review of "Use of digital strategies in the diagnosis of oral squamous cell carcinoma: a scoping review"

_PeerJ, doi:10.7717/peerj.17329_

## Round 0.1 · original submission · Major Revisions

Please respond to all queries in detail.

**Language Note:** The review process has identified that the English language must be improved. PeerJ can provide language editing services - please contact us at [email protected] for pricing (be sure to provide your manuscript number and title). Alternatively, you should make your own arrangements to improve the language quality and provide details in your response letter. – PeerJ Staff

Reviewer 1 ·

Basic reporting

Nice review article of the world literature on the current status of digital strategies used in the remote diagnosis of Oral Squamous Cell Carcinoma [OSCC].

Experimental design

Extensive assessment of the published literature from a broad spectrum of platforms, including PubMed/MEDLINE, Scopus, Web of Science, Embase and Sciencedirect. Encompasses the countries where the remote diagnosis is of day-to-day practical importance, especially in India and Brazil, where large distances and accessibility considerations make remote consultations a necessity.
The literature reviewed is stated as including studies in English, but 3 of the 33 references are in a language other than English [Portuguese]: References 2, 3 and 30. If these references have abstracts in English, it is acceptable to keep them.
The search strategies and tables are valuable in providing details of the literature analyzed.

Validity of the findings

The analysis of the data from the relevant literature is carried out correctly throughout the paper. Some of the granular details provide a window on the diagnostic accuracy in identifying OSCC in various parts of the world.
One wonders why dentistry students [who are not yet certified specialists in their field] would have been included in providing diagnostic opinions on oral lesions? This fact is then reflected in the higher rate of diagnostic accuracy from the certified specialist in oral medicine/dentistry.

Additional comments

Nice addition to the literature on the topic of remote diagnosis of oral lesions using a variety of telediagnosis tools across the world. By its design the review does not present new material, but it is nonetheless a state-of-the-art snapshot in time of the relevant literature at the time of the study [June 2023].
In Table 1, it may not be necessary to include a separate column for the date, since all the searches were carried out on the same day [June 18, 2023]. Instead, it might suffice to state as an additional line in the title or in the paragraph title (‘Search strategies’): ‘All searches as of June 18, 2023.', or some similar statement.
The English used in the paper is of very high quality. However, some sentences seem to be constructed along idioms common in another language [Portuguese, one would presume].
For example, line 212: "The present study observed the TD proved to be feasible in the interpretation…". Instead, a more English-sounding wording might be: 'The present study concluded that TD is a practical tool in the interpretation…'.
Or, line 234: "It presents itself as an alternative…". Instead, 'It represents an alternative…' sounds more in accordance with accepted English usage.

There are a few typos in the text.
Line 116: "The figure 2"; instead: 'Figure 2" [delete ‘The’].
Line 117: 'Figure 3…', and 'Figure 4…' [delete ‘The’].
Line 123: '… 8,516 patients. Of whom…" is incorrect in English. Instead, either '… 8,516 patients. Of these…', or '...8,516 patients, of whom…'.
Line 257: "… number of publications [delete "than"] Senegal…”. Instead, '… number of publications from Senegal…’.
Line 266: "…which may lead to…”; delete the word ‘which’.
Line 270: “…studies even if the third reviewer…”; correctly: ‘…studies even if a third reviewer…’.

Reviewer 2 ·

Basic reporting

.

Experimental design

.

Validity of the findings

.

Additional comments

Please find the reviewed pdf attached. Comments are highlighted.

Overall, it is an average review article with below average grammar and occasionally ambiguous language.
However, the research question and investigations are relevant and conclusion is also in line with them.

The ethical aspect of privacy of patient identity and data in teleconsultations needs to be addressed or at least mentioned somewhere.

Annotated reviews are not available for download in order to protect the identity of reviewers who chose to remain anonymous.

---

## Round 0.2 · accepted · Accept

Congratulations on this remarkable achievement!

Reviewer 1 ·

Basic reporting

OK after corrections.

Experimental design

OK after corrections.

Validity of the findings

OK after corrections.

Additional comments

OK after corrections.

Reviewer 2 ·

Basic reporting

.

Experimental design

.

Validity of the findings

.

Additional comments

I have reviewed article again, no significant changes are required. It’s okay from my side.